# Learning Long-Term Reward Redistribution via Randomized Return Decomposition

**Zhizhou Ren**[1], **Ruihan Guo**[2], **Yuan Zhou**[3], **Jian Peng**[145]
[1]University of Illinois at Urbana-Champaign  [2]Shanghai Jiao Tong University
[3]BIMSA  [4]AIR, Tsinghua University  [5]HeliXon Limited
`zhizhour@illinois.edu, guoruihan@sjtu.edu.cn`
`timzhouyuan@gmail.com, jianpeng@illinois.edu`

## Abstract

Many practical applications of reinforcement learning require agents to learn from sparse and delayed rewards. It challenges the ability of agents to attribute their actions to future outcomes. In this paper, we consider the problem formulation of episodic reinforcement learning with trajectory feedback. It refers to an extreme delay of reward signals, in which the agent can only obtain one reward signal at the end of each trajectory. A popular paradigm for this problem setting is learning with a designed auxiliary dense reward function, namely proxy reward, instead of sparse environmental signals. Based on this framework, this paper proposes a novel reward redistribution algorithm, randomized return decomposition (RRD), to learn a proxy reward function for episodic reinforcement learning. We establish a surrogate problem by Monte-Carlo sampling that scales up least-squares-based reward redistribution to long-horizon problems. We analyze our surrogate loss function by connection with existing methods in the literature, which illustrates the algorithmic properties of our approach. In experiments, we extensively evaluate our proposed method on a variety of benchmark tasks with episodic rewards and demonstrate substantial improvement over baseline algorithms.

## 1 Introduction

Scaling reinforcement learning (RL) algorithms to practical applications has become the focus of numerous recent studies, including resource management (Mao et al., 2016), industrial control (Hein et al., 2017), drug discovery (Popova et al., 2018), and recommendation systems (Chen et al., 2018). One of the challenges in these real-world problems is the sparse and delayed environmental rewards. For example, in the molecular structure design problem, the target molecule property can only be evaluated after completing the whole sequence of modification operations (Zhou et al., 2019b). The sparsity of environmental feedback would complicate the attribution of rewards on agent actions and therefore can hinder the efficiency of learning (Rahmandad et al., 2009). In practice, it is a common choice to formulate the RL objective with a meticulously designed reward function instead of the sparse environmental rewards. The design of such a reward function is crucial to the performance of the learned policies. Most standard RL algorithms, such as temporal difference learning and policy gradient methods, prefer dense reward functions that can provide instant feedback for every step of environment transitions. Designing such dense reward functions is not a simple problem even with domain knowledge and human supervision. It has been widely observed in prior works that handcrafted heuristic reward functions may lead to unexpected and undesired behaviors (Randløv & Alstrøm, 1998; Bottou et al., 2013; Andrychowicz et al., 2017). The agent may find a shortcut solution that formally optimizes the given objective but deviates from the desired policies (Dewey, 2014; Amodei et al., 2016). The reward designer can hardly anticipate all potential side effects of the designed reward function, which highlights the difficulty of reward engineering.

To avoid the unintended behaviors induced by misspecified reward engineering, a common paradigm is considering the reward design as an online problem within the trial-and-error loop of reinforcement learning (Sorg et al., 2010). This algorithmic framework contains two components, namely reward modeling and policy optimization. The agent first learns a proxy reward function from the experience data and then optimizes its policy based on the learned per-step rewards. By iterating this

procedure and interacting with the environment, the agent is able to continuously refine its reward model so that the learned proxy reward function can better approximate the actual objective given by the environmental feedback. More specifically, this paradigm aims to reshape the sparse and delayed environmental rewards to a dense Markovian reward function while trying to avoid misspecifying the goal of given tasks.

In this paper, we propose a novel reward redistribution algorithm based on a classical mechanism called *return decomposition* (Arjona-Medina et al., 2019). Our method is built upon the least-squares-based return decomposition (Efroni et al., 2021) whose basic idea is training a regression model that decomposes the trajectory return to the summation of per-step proxy rewards. This paradigm is a promising approach to redistributing sparse environmental feedback. Our proposed algorithm, *randomized return decomposition* (RRD), establish a surrogate optimization of return decomposition to improve the scalability in long-horizon tasks. In this surrogate problem, the reward model is trained to predict the episodic return from a random subsequence of the agent trajectory, i.e., we conduct a structural constraint that the learned proxy rewards can approximately reconstruct environmental trajectory return from a small subset of state-action pairs. This design enables us to conduct return decomposition effectively by mini-batch training. Our analysis shows that our surrogate loss function is an upper bound of the original loss of deterministic return decomposition, which gives a theoretical interpretation of this randomized implementation. We also present how the surrogate gap can be controlled and draw connections to another method called uniform reward redistribution. In experiments, we demonstrate substantial improvement of our proposed approach over baseline algorithms on a suite of MuJoCo benchmark tasks with episodic rewards.

## 2 BACKGROUND

### 2.1 EPISODIC REINFORCEMENT LEARNING WITH TRAJECTORY FEEDBACK

In standard reinforcement learning settings, the environment model is usually formulated by a *Markov decision process* (MDP; Bellman, 1957), defined as a tuple $\mathcal{M} = \langle \mathcal{S}, \mathcal{A}, P, R, \mu \rangle$, where $\mathcal{S}$ and $\mathcal{A}$ denote the spaces of environment states and agent actions. $P(s'|s, a)$ and $R(s, a)$ denote the unknown environment transition and reward functions. $\mu$ denotes the initial state distribution. The goal of reinforcement learning is to find a policy $\pi : \mathcal{S} \to \mathcal{A}$ maximizing cumulative rewards. More specifically, a common objective is maximizing infinite-horizon discounted rewards based on a pre-defined discount factor $\gamma$ as follows:

$$\text{(standard objective)} \quad J(\pi) = \mathbb{E}\left[ \sum_{t=0}^{\infty} \gamma^t R(s_t, \pi(s_t)) \;\middle|\; s_0 \sim \mu, \; s_{t+1} \sim P(\cdot \mid s_t, \pi(s_t)) \right]. \quad (1)$$

In this paper, we consider the episodic reinforcement learning setting with trajectory feedback, in which the agent can only obtain one reward feedback at the end of each trajectory. Let $\tau$ denote an agent trajectory that contains all experienced states and behaved actions within an episode. We assume all trajectories terminate in finite steps. The episodic reward function $R_{\text{ep}}(\tau)$ is defined on the trajectory space, which represents the overall performance of trajectory $\tau$. The goal of episodic reinforcement learning is to maximize the expected trajectory return:

$$\text{(episodic objective)} \quad J_{\text{ep}}(\pi) = \mathbb{E}\left[ R_{\text{ep}}(\tau) \;\middle|\; s_0 \sim \mu, \; a_t = \pi(s_t), \; \tau = \langle s_0, a_0, s_1, \cdots, s_T \rangle \right]. \quad (2)$$

In general, the episodic-reward setting is a particular form of *partially observable Markov decision processes* (POMDPs) where the reward function is non-Markovian. The worst case may require the agent to enumerate the entire exponential-size trajectory space for recovering the episodic reward function. In practical problems, the episodic environmental feedback usually has structured representations. A common structural assumption is the existence of an underlying Markovian reward function $\widehat{R}(s, a)$ that approximates the episodic reward $R_{\text{ep}}(\tau)$ by a sum-form decomposition,

$$\text{(sum-decomposable episodic reward)} \quad R_{\text{ep}}(\tau) \approx \widehat{R}_{\text{ep}}(\tau) = \sum_{t=0}^{T-1} \widehat{R}(s_t, a_t). \quad (3)$$

This structure is commonly considered by both theoretical (Efroni et al., 2021) and empirical studies (Liu et al., 2019; Raposo et al., 2021) on long-horizon episodic rewards. It models the situations

where the agent objective is measured by some metric with additivity properties, e.g., the distance of robot running, the time cost of navigation, or the number of products produced in a time interval.

## 2.2 Reward Redistribution

The goal of *reward redistribution* is constructing a proxy reward function $\widehat{R}(s_t, a_t)$ that transforms the episodic-reward problem stated in Eq. (2) to a standard dense-reward setting. By replacing environmental rewards with such a Markovian proxy reward function $\widehat{R}(s_t, a_t)$, the agent can be trained to optimize the discounted objective in Eq. (1) using any standard RL algorithms. Formally, the proxy rewards $\widehat{R}(s_t, a_t)$ form a sum-decomposable reward function $\widehat{R}_{\mathrm{ep}}(\tau) = \sum_{t=0}^{T-1} \widehat{R}(s_t, a_t)$ that is expected to have high correlation to the environmental reward $R_{\mathrm{ep}}(\tau)$. Here, we introduce two branches of existing reward redistribution methods, *return decomposition* and *uniform reward redistribution*, which are the most related to our proposed approach. We defer the discussions of other related work to section 5.

**Return Decomposition.** The idea of *return decomposition* is training a reward model that predicts the trajectory return with a given state-action sequence (Arjona-Medina et al., 2019). In this paper, without further specification, we focus on the least-squares-based implementation of return decomposition (Efroni et al., 2021). The reward redistribution is given by the learned reward model, i.e., decomposing the environmental episodic reward $R_{\mathrm{ep}}(\tau)$ to a Markovian proxy reward function $\widehat{R}(s, a)$. In practice, the reward modeling is formulated by optimizing the following loss function:

$$\mathcal{L}_{\mathrm{RD}}(\theta) = \mathop{\mathbb{E}}_{\tau \sim \mathcal{D}} \left[ \left( R_{\mathrm{ep}}(\tau) - \sum_{t=0}^{T-1} \widehat{R}_\theta(s_t, a_t) \right)^2 \right], \tag{4}$$

where $\widehat{R}_\theta$ denotes the parameterized proxy reward function, $\theta$ denotes the parameters of the learned reward model, and $\mathcal{D}$ denotes the experience dataset collected by the agent. Assuming the sum-decomposable structure stated in Eq. (3), $\widehat{R}_\theta(s, a)$ is expected to asymptotically concentrate near the ground-truth underlying rewards $\widehat{R}(s, a)$ when Eq. (4) is properly optimized (Efroni et al., 2021).

One limitation of the least-squares-based return decomposition method specified by Eq. (4) is its scalability in terms of the computation costs. Note that the trajectory-wise episodic reward is the only environmental supervision for reward modeling. Computing the loss function $\mathcal{L}_{\mathrm{RD}}(\theta)$ with a single episodic reward label requires to enumerate all state-action pairs along the whole trajectory. This computation procedure can be expensive in numerous situations, e.g., when the task horizon $T$ is quite long, or the state space $\mathcal{S}$ is high-dimensional. To address this practical barrier, recent works focus on designing reward redistribution mechanisms that can be easily integrated in complex tasks. We will discuss the implementation subtlety of existing methods in section 4.

**Uniform Reward Redistribution.** To pursue a simple but effective reward redistribution mechanism, IRCR (Gangwani et al., 2020) considers *uniform reward redistribution* which assumes all state-action pairs equally contribute to the return value. It is designed to redistribute rewards in the absence of any prior structure or information. More specifically, the proxy reward $\widehat{R}_{\mathrm{IRCR}}(s, a)$ is computed by averaging episodic return values over all experienced trajectories containing $(s, a)$,

$$\widehat{R}_{\mathrm{IRCR}}(s, a) = \mathop{\mathbb{E}}_{\tau \sim \mathcal{D}} \big[ R_{\mathrm{ep}}(\tau) \mid (s, a) \in \tau \big]. \tag{5}$$

In this paper, we will introduce a novel reward redistribution mechanism that bridges between return decomposition and uniform reward redistribution.

## 3 Reward Redistribution via Randomized Return Decomposition

In this section, we introduce our approach, randomized return decomposition (RRD), which sets up a surrogate optimization problem of the least-squares-based return decomposition. The proposed surrogate objective allows us to conduct return decomposition on short subsequences of agent trajectories, which is scalable in long-horizon tasks. We provide analyses to characterize the algorithmic property of our surrogate objective function and discuss connections to existing methods.

### 3.1 Randomized Return Decomposition with Monte-Carlo Return Estimation

One practical barrier to apply least-squares-based return decomposition methods in long-horizon tasks is the computation costs of the regression loss in Eq. (4), i.e., it requires to enumerate all state-action pairs within the agent trajectory. To resolve this issue, we consider a randomized method that uses a Monte-Carlo estimator to compute the predicted episodic return $\widehat{R}_{\mathrm{ep},\theta}(\tau)$ as follows:

$$\underbrace{\widehat{R}_{\mathrm{ep},\theta}(\tau) = \sum_{t=0}^{T-1} \widehat{R}_\theta(s_t, a_t)}_{\text{Deterministic Computation}} = \mathop{\mathbb{E}}_{\mathcal{I} \sim \rho_T(\cdot)}\left[\frac{T}{|\mathcal{I}|}\sum_{t \in \mathcal{I}} \widehat{R}_\theta(s_t, a_t)\right] \approx \underbrace{\frac{T}{|\mathcal{I}|}\sum_{t \in \mathcal{I}} \widehat{R}_\theta(s_t, a_t)}_{\text{Monte-Carlo Estimation}}, \tag{6}$$

where $\mathcal{I}$ denotes a subset of indices. $\rho_T(\cdot)$ denotes an unbiased sampling distribution where each index $t$ has the same probability to be included in $\mathcal{I}$. In this paper, without further specification, $\rho_T(\cdot)$ is constructed by uniformly sampling $K$ distinct indices.

$$\rho_T(\cdot) = \mathrm{Uniform}\left(\left\{\mathcal{I} \subseteq \mathbb{Z}_T : |\mathcal{I}| = K\right\}\right), \tag{7}$$

where $K$ is a hyper-parameter. In this sampling distribution, each timestep $t$ has the same probability to be covered by the sampled subsequence $\mathcal{I} \sim \rho_T(\cdot)$ so that it gives an unbiased Monte-Carlo estimation of the episodic summation $\widehat{R}_{\mathrm{ep},\theta}(\tau)$.

**Randomized Return Decomposition.** Based on the idea of using Monte-Carlo estimation shown in Eq. (6), we introduce our approach, *randomized return decomposition* (RRD), to improve the scalability of least-squares-based reward redistribution methods. The objective function of our approach is formulated by the randomized return decomposition loss $\mathcal{L}_{\mathrm{Rand\text{-}RD}}(\theta)$ stated in Eq. (8), in which the parameterized proxy reward function $\widehat{R}_\theta$ is trained to predict the episodic return $R_{\mathrm{ep}}(\tau)$ given a random subsequence of the agent trajectory. In other words, we integrate the Monte-Carlo estimator (see Eq. (6)) into the return decomposition loss to obtain the following surrogate loss function:

$$\mathcal{L}_{\mathrm{Rand\text{-}RD}}(\theta) = \mathop{\mathbb{E}}_{\tau \sim \mathcal{D}}\left[\mathop{\mathbb{E}}_{\mathcal{I} \sim \rho_T(\cdot)}\left[\left(R_{\mathrm{ep}}(\tau) - \frac{T}{|\mathcal{I}|}\sum_{t \in \mathcal{I}} \widehat{R}_\theta(s_t, a_t)\right)^2\right]\right]. \tag{8}$$

In practice, the loss function $\mathcal{L}_{\mathrm{Rand\text{-}RD}}(\theta)$ can be estimated by sampling a mini-batch of trajectory subsequences instead of computing $\widehat{R}_\theta(s_t, a_t)$ for the whole agent trajectory, and thus the implementation of randomized return decomposition is adaptive and flexible in long-horizon tasks.

### 3.2 Analysis of Randomized Return Decomposition

The main purpose of our approach is establishing a surrogate loss function to improve the scalability of least-squares-based return decomposition in practice. Our proposed method, randomized return decomposition, is a trade-off between the computation complexity and the estimation error induced by the Monte-Carlo estimator. In this section, we show that our approach is an interpolation between between the return decomposition paradigm and uniform reward redistribution, which can be controlled by the hyper-parameter $K$ used in the sampling distribution (see Eq. (7)). We present Theorem 1 as a formal characterization of our proposed surrogate objective function.

**Theorem 1** (Loss Decomposition). *The surrogate loss function $\mathcal{L}_{Rand\text{-}RD}(\theta)$ can be decomposed to two terms interpolating between return decomposition and uniform reward redistribution.*

$$\mathcal{L}_{Rand\text{-}RD}(\theta) = \mathcal{L}_{RD}(\theta) + \mathop{\mathbb{E}}_{\tau \sim \mathcal{D}}\left[\underbrace{\mathop{Var}_{\mathcal{I} \sim \rho_T(\cdot)}\left[\frac{T}{|\mathcal{I}|}\sum_{t \in \mathcal{I}} \widehat{R}_\theta(s_t, a_t)\right]}_{\textit{variance of the Monte-Carlo estimator}}\right] \tag{9}$$

$$= \underbrace{\mathcal{L}_{RD}(\theta)}_{\textit{return decomposition}} + \mathop{\mathbb{E}}_{\tau \sim \mathcal{D}}\left[T^2 \cdot \underbrace{\mathop{Var}_{(s_t, a_t) \sim \tau}\left[\widehat{R}_\theta(s_t, a_t)\right]}_{\textit{uniform reward redistribution}} \cdot \underbrace{\frac{1}{K}\left(1 - \frac{K-1}{T-1}\right)}_{\textit{interpolation weight}}\right], \tag{10}$$

*where $K$ denotes the length of sampled subsequences defined in Eq. (7).*

The proof of Theorem 1 is based on the bias-variance decomposition formula of mean squared error (Kohavi & Wolpert, 1996). The detailed proofs are deferred to Appendix A.

**Interpretation as Regularization.** As presented in Theorem 1, the randomized decomposition loss can be decomposed to two terms, the deterministic return decomposition loss $\mathcal{L}_{\text{RD}}(\theta)$ and a variance penalty term (see the second term of Eq. (9)). The variance penalty term can be regarded as a regularization that is controlled by hyper-parameter $K$. In practical problems, the objective of return decomposition is ill-posed, since the number of trajectory labels is dramatically less than the number of transition samples. There may exist solutions of reward redistribution that formally optimize the least-squares regression loss but serve little functionality to guide the policy learning. Regarding this issue, the variance penalty in randomized return decomposition is a regularizer for reward modeling. It searches for smooth proxy rewards that has low variance within the trajectory. This regularization effect is similar to the mechanism of uniform reward redistribution (Gangwani et al., 2020), which achieves state-of-the-art performance in the previous literature. In section 4, our experiments demonstrate that the variance penalty is crucial to the empirical performance of randomized return decomposition.

**A Closer Look at Loss Decomposition.** In addition to the intuitive interpretation of regularization, we will present a detailed characterization of the loss decomposition shown in Theorem 1. We interpret this loss decomposition as below:

1. Note that the Monte-Carlo estimator used by randomized return decomposition is an unbiased estimation of the proxy episodic return $\widehat{R}_{\text{ep},\theta}(\tau)$ (see Eq. (6)). This unbiased property gives the first component of the loss decomposition, i.e, the original return decomposition loss $\mathcal{L}_{\text{RD}}(\theta)$.

2. Although the Monte-Carlo estimator is unbiased, its variance would contribute to an additional loss term induced by the mean-square operator, i.e., the second component of loss decomposition presented in Eq. (10). This additional term penalizes the variance of the learned proxy rewards under random sampling. This penalty expresses the same mechanism as uniform reward redistribution (Gangwani et al., 2020) in which the episodic return is uniformly redistributed to the state-action pairs in the trajectory.

Based on the above discussions, we can analyze the algorithmic properties of randomized return decomposition by connecting with previous studies.

### 3.2.1 SURROGATE OPTIMIZATION OF RETURN DECOMPOSITION

Randomized return decomposition conducts a surrogate optimization of the actual return decomposition. Note that the variance penalty term in Eq. (10) is non-negative, our loss function $\mathcal{L}_{\text{Rand-RD}}(\theta)$ serves an upper bound estimation of the original loss $\mathcal{L}_{\text{RD}}(\theta)$ as the following statement.

**Proposition 1** (Surrogate Upper Bound). *The randomized return decomposition loss $\mathcal{L}_{Rand\text{-}RD}(\theta)$ is an upper bound of the actual return decomposition loss function $\mathcal{L}_{RD}(\theta)$, i.e., $\mathcal{L}_{Rand\text{-}RD}(\theta) \geq \mathcal{L}_{RD}(\theta)$.*

Proposition 1 suggests that optimizing our surrogate loss $\mathcal{L}_{\text{Rand-RD}}(\theta)$ guarantees to optimize an upper bound of the actual return decomposition loss $\mathcal{L}_{\text{RD}}(\theta)$. According to Theorem 1, the gap between $\mathcal{L}_{\text{Rand-RD}}(\theta)$ and $\mathcal{L}_{\text{RD}}(\theta)$ refers to the variance of subsequence sampling. The magnitude of this gap can be controlled by the hyper-parameter $K$ that refers to the length of sampled subsequences.

**Proposition 2** (Objective Gap). *Let $\mathcal{L}_{Rand\text{-}RD}^{(K)}(\theta)$ denote the randomized return decomposition loss that samples subsequences with length $K$. The gap between $\mathcal{L}_{Rand\text{-}RD}^{(K)}(\theta)$ and $\mathcal{L}_{RD}(\theta)$ can be reduced by using larger values of hyper-parameter $K$.*

$$\forall \theta, \quad \mathcal{L}_{Rand\text{-}RD}^{(1)}(\theta) \geq \mathcal{L}_{Rand\text{-}RD}^{(2)}(\theta) \geq \cdots \geq \mathcal{L}_{Rand\text{-}RD}^{(T-1)}(\theta) \geq \mathcal{L}_{Rand\text{-}RD}^{(T)}(\theta) = \mathcal{L}_{RD}(\theta). \quad (11)$$

This gap can be eliminated by choosing $K = T$ in the sampling distribution (see Eq. (7)) so that our approach degrades to the original deterministic implementation of return decomposition.

### 3.2.2 GENERALIZATION OF UNIFORM REWARD REDISTRIBUTION

The reward redistribution mechanism of randomized return decomposition is a generalization of uniform reward redistribution. To serve intuitions, we start the discussions with the simplest case using $K = 1$ in subsequence sampling, in which our approach degrades to the uniform reward redistribution as the following statement.

**Proposition 3** (Uniform Reward Redistribution). *Assume all trajectories have the same length and the parameterization space of $\theta$ serves universal representation capacity. The optimal solution $\theta^\star$ of minimizing $\mathcal{L}_{Rand\text{-}RD}^{(1)}(\theta)$ is stated as follows:*

$$\widehat{R}_{\theta^\star}(s,a) = \mathop{\mathbb{E}}_{\tau \sim \mathcal{D}} \big[ R_{ep}(\tau)/T \mid (s,a) \in \tau \big], \tag{12}$$

*where $\mathcal{L}_{Rand\text{-}RD}^{(1)}(\theta)$ denotes the randomized return decomposition loss with $K = 1$.*

A minor difference between Eq. (12) and the proxy reward designed by Gangwani et al. (2020) (see Eq. (5)) is a multiplier scalar $1/T$. Gangwani et al. (2020) interprets such a proxy reward mechanism as a trajectory-space smoothing process or a non-committal reward redistribution. Our analysis can give a mathematical characterization to illustrate the objective of uniform reward redistribution. As characterized by Theorem 1, uniform reward redistribution conducts an additional regularizer to penalize the variance of per-step proxy rewards. In the view of randomized return decomposition, the functionality of this regularizer is requiring the reward model to reconstruct episodic return from each single-step transition.

By using larger values of hyper-parameter $K$, randomized return decomposition is trained to reconstruct the episodic return from a subsequence of agent trajectory instead of the single-step transition used by uniform reward redistribution. This mechanism is a generalization of uniform reward redistribution, in which we equally assign rewards to subsequences generated by uniformly random sampling. It relies on the concentratability of random sampling, i.e., the average of a sequence can be estimated by a small random subset with sub-linear size. The individual contribution of each transition within the subsequence is further attributed by return decomposition.

### 3.3 PRACTICAL IMPLEMENTATION OF RANDOMIZED RETURN DECOMPOSITION

In Algorithm 1, we integrate randomized return decomposition with policy optimization. It follows an iterative paradigm that iterates between the rewarding modeling and policy optimization modules.

---

**Algorithm 1** Policy Optimization with Randomized Return Decomposition

---

1: Initialize $\mathcal{D} \leftarrow \emptyset$
2: **for** $\ell = 1, 2, \cdots$ **do**
3:      Collect a rollout trajectory $\tau$ using the current policy.
4:      Store trajectory $\tau$ and feedback $R_{ep}(\tau)$ into the replay buffer $\mathcal{D} \leftarrow \mathcal{D} \cup \{(\tau, R_{ep}(\tau))\}$.
5:      **for** $i = 1, 2, \cdots$ **do**
6:          Sample $M$ trajectories $\{\tau_j \in \mathcal{D}\}_{j=1}^{M}$ from the replay buffer.
7:          Sample subsequences $\{\mathcal{I}_j \subseteq \mathbb{Z}_{T_j}\}_{j=1}^{M}$ for these trajectories.
8:          Estimate randomized return decomposition loss $\widehat{\mathcal{L}}_{\text{Rand-RD}}(\theta)$,

$$\widehat{\mathcal{L}}_{\text{Rand-RD}}(\theta) = \frac{1}{M} \sum_{j=1}^{M} \left[ \left( R_{ep}(\tau_j) - \frac{T_j}{|\mathcal{I}_j|} \sum_{t \in \mathcal{I}_j} \widehat{R}_\theta(s_{j,t}, a_{j,t}) \right)^2 \right], \tag{13}$$

         where $T_j$ denotes the length of trajectory $\tau_j = \langle s_{j,1}, a_{j,1}, \cdots, s_{j,T_j} \rangle$.
9:          Perform a gradient update on the reward model $\widehat{R}_\theta$,

$$\theta \leftarrow \theta - \alpha \nabla_\theta \widehat{\mathcal{L}}_{\text{Rand-RD}}(\theta), \tag{14}$$

         where $\alpha$ denotes the learning rate.
10:     Perform policy optimization using the learned proxy reward function $\widehat{R}_\theta(s, a)$.

---

As presented in Eq. (13) and Eq. (14), the optimization of our loss function $\mathcal{L}_{\text{Rand-RD}}(\theta)$ can be easily conducted by mini-batch gradient descent. This surrogate loss function only requires computations on short-length subsequences. It provides a scalable implementation for return decomposition that can be generalized to long-horizon tasks with manageable computation costs. In section 4, we will show that this simple implementation can also achieve state-of-the-art performance in comparison to other existing methods.

## 4  EXPERIMENTS

In this section, we investigate the empirical performance of our proposed methods by conducting experiments on a suite of MuJoCo benchmark tasks with episodic rewards. We compare our approach with several baseline algorithms in the literature and conduct an ablation study on subsequence sampling that is the core component of our algorithm.

### 4.1  PERFORMANCE EVALUATION ON MUJOCO BENCHMARK WITH EPISODIC REWARDS

**Experiment Setting.** We adopt the same experiment setting as Gangwani et al. (2020) to compare the performance of our approach with baseline algorithms. The experiment environments is based on the MuJoCo locomotion benchmark tasks created by OpenAI Gym (Brockman et al., 2016). These tasks are long-horizon with maximum trajectory length $T = 1000$, i.e., the task horizon is definitely longer than the batch size used by the standard implementation of mini-batch gradient estimation. We modify the reward function of these environments to set up an episodic-reward setting. Formally, on non-terminal states, the agent will receive a zero signal instead of the per-step dense rewards. The agent can obtain the episodic feedback $R_{ep}(\tau)$ at the last step of the rollout trajectory, in which $R_{ep}(\tau)$ is computed by the summation of per-step instant rewards given by the standard setting. We evaluate the performance of our proposed methods with the same configuration of hyper-parameters in all environments. A detailed description of implementation details is included in Appendix B.

We evaluate two implementations of randomized return decomposition (RRD):

- **RRD (ours)** denotes the default implementation of our approach. We train a reward model using randomized return decomposition loss $\mathcal{L}_{\text{Rand-RD}}$, in which we sample subsequences with length $K = 64$ in comparison to the task horizon $T = 1000$. The reward model $\widehat{R}_\theta$ is parameterized by a two-layer fully connected network. The policy optimization module is implemented by soft actor-critic (SAC; Haarnoja et al., 2018a). [1]

- **RRD-$\mathcal{L}_{\text{RD}}$ (ours)** is an alternative implementation that optimizes $\mathcal{L}_{\text{RD}}$ instead of $\mathcal{L}_{\text{Rand-RD}}$. Note that Theorem 1 gives a closed-form characterization of the gap between $\mathcal{L}_{\text{Rand-RD}}(\theta)$ and $\mathcal{L}_{\text{RD}}(\theta)$, which is represented by the variance of the learned proxy rewards. By subtracting an unbiased variance estimation from loss function $\mathcal{L}_{\text{Rand-RD}}(\theta)$, we can estimate loss function $\mathcal{L}_{\text{RD}}(\theta)$ by sampling short subsequences. It gives a computationally efficient way to optimize $\mathcal{L}_{\text{RD}}(\theta)$. We include this alternative implementation to reveal the functionality of the regularization given by variance penalty. A detailed description is deferred to Appendix B.3.

We compare with several existing methods for episodic or delayed reward settings:

- **IRCR** (Gangwani et al., 2020) is an implementation of uniform reward redistribution. The reward redistribution mechanism of IRCR is non-parametric, in which the proxy reward of a transition is set to be the normalized value of corresponding trajectory return. It is equivalent to use a Monte-Carlo estimator of Eq. (5). Due to the ease of implementation, this method achieves state-of-the-art performance in the literature.

- **RUDDER** (Arjona-Medina et al., 2019) is based on the idea of return decomposition but does not directly optimize $\mathcal{L}_{\text{RD}}(\theta)$. Instead, it trains a return predictor based on trajectory, and the step-wise credit is assigned by the prediction difference between two consecutive states. By using the warm-up technique of LSTM, this transform prevents its training computation costs from depending on the task horizon $T$ so that it is adaptive to long-horizon tasks.

- **GASIL** (Guo et al., 2018), generative adversarial self-imitation learning, is a generalization of GAIL (Ho & Ermon, 2016). It formulates an imitation learning framework by imitating best trajectories in the replay buffer. The proxy rewards are given by a discriminator that is trained to classify the agent and expert trajectories.

- **LIRPG** (Zheng et al., 2018) aims to learn an intrinsic reward function from sparse environment feedback. Its policy is trained to optimized the sum of the extrinsic and intrinsic rewards. The parametric intrinsic reward function is updated by meta-gradients to optimize the actual extrinsic rewards achieved by the policy

---

[1]The source code of our implementation is available at `https://github.com/Stilwell-Git/Randomized-Return-Decomposition`.

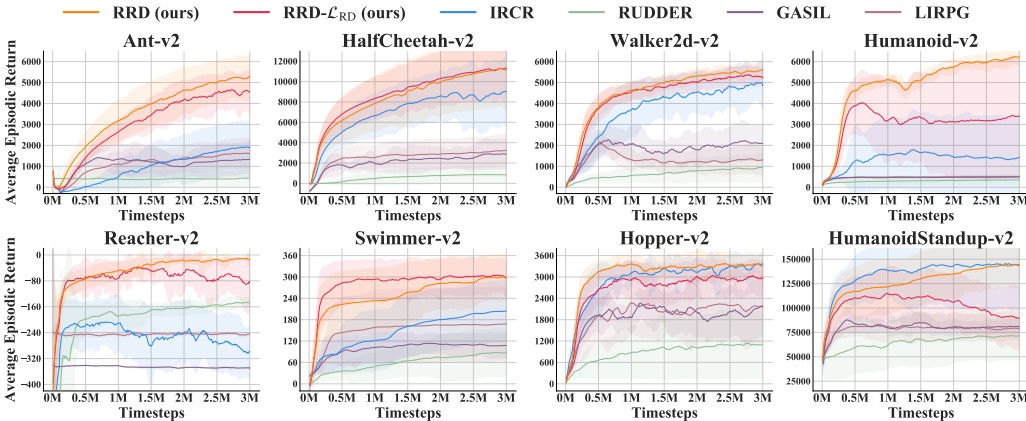

Figure 1: Learning curves on a suite of MuJoCo benchmark tasks with episodic rewards. All curves for MuJoCo benchmark are plotted from 30 runs with random initializations. The shaded region indicates the standard deviation. To make the comparison more clear, the curves are smoothed by averaging 10 most recent evaluation points. We set up an evaluation point every $10^4$ timesteps.

**Overall Performance Comparison.** As presented in Figure 1, randomized return decomposition generally outperforms baseline algorithms. Our approach can achieve higher sample efficiency and produce better policies after convergence. RUDDER is an implementation of return decomposition that represents single-step rewards by the differences between the return predictions of two consecutive states. This implementation maintains high computation efficiency but long-term return prediction is a hard optimization problem and requires on-policy samples. In comparison, RRD is a more scalable and stable implementation which can better integrate with off-policy learning for improving sample efficiency. The uniform reward redistribution considered by IRCR is simple to implement but cannot extract the temporal structure of episodic rewards. Thus the final policy quality produced by RRD is usually better than that of IRCR. GASIL and LIRPG aim to construct auxiliary reward functions that have high correlation to the environmental return. These two methods cannot achieve high sample efficiency since their objectives require on-policy training.

**Variance Penalty as Regularization.** Figure 1 also compares two implementations of randomized return decomposition. In most testing environments, RRD optimizing $\mathcal{L}_{\text{Rand-RD}}$ outperforms the unbiased implementation RRD-$\mathcal{L}_{\text{RD}}$. We consider RRD using $\mathcal{L}_{\text{Rand-RD}}$ as our default implementation since it performs better and its objective function is simpler to implement. As discussed in section 3.2, the variance penalty conducted by RRD aims to minimize the variance of the Monte-Carlo estimator presented in Eq. (6). It serves as a regularization to restrict the solution space of return decomposition, which gives two potential effects: (1) RRD prefers smooth proxy rewards when the expressiveness capacity of reward network over-parameterizes the dataset. (2) The variance of mini-batch gradient estimation can also be reduced when the variance of Monte-Carlo estimator is small. In practice, this regularization would benefit the training stability. As presented in Figure 1, RRD achieves higher sample efficiency than RRD-$\mathcal{L}_{\text{RD}}$ in most testing environments. The quality of the learned policy of RRD is also better than that of RRD-$\mathcal{L}_{\text{RD}}$. It suggests that the regularized reward redistribution can better approximate the actual environmental objective.

## 4.2 ABLATION STUDIES

We conduct an ablation study on the hyper-parameter $K$ that represent the length of subsequences used by randomized return decomposition. As discussed in section 3.2, the hyper-parameter $K$ controls the interpolation ratio between return decomposition and uniform reward redistribution. It trades off the accuracy of return reconstruction and variance regularization. In Figure 2, we evaluate RRD with a set of choices of hyper-parameter $K \in \{1, 8, 16, 32, 64, 128\}$. The experiment results show that, although the sensitivity of this hyper-parameter depends on the environment, a relatively large value of $K$ generally achieves better performance, since it can better approximate the environmental objective. In this experiment, we ensure all runs use the same input size for mini-batch training, i.e., using larger value of $K$ leads to less number of subsequences in the mini-batch.

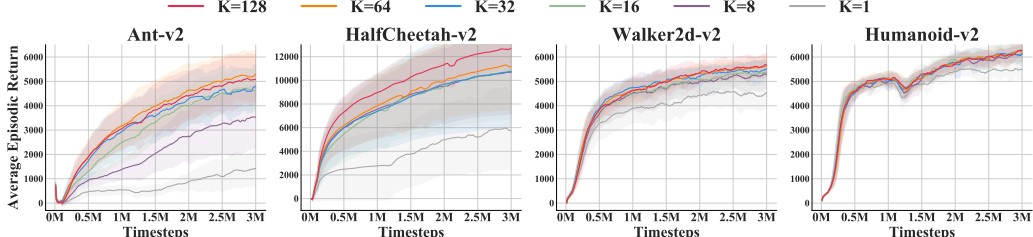

Figure 2: Learning curves of RRD with different choices of hyper-parameter $K$. The curves with $K = 64$ correspond to the default implementation of RRD presented in Figure 1.

More specifically, in this ablation study, all algorithm instances estimate the loss function $\mathcal{L}_{\text{Rand-RD}}$ using a mini-batch containing 256 transitions. We consider $K = 64$ as our default configuration. As presented in Figure 2, larger values give marginal improvement in most environments. The benefits of larger values of $K$ can only be observed in HalfCheetah-v2. We note that HalfCheetah-v2 does not have early termination and thus has the longest average horizon among these locomotion tasks. It highlights the trade-off between the weight of regularization and the bias of subsequence estimation.

## 5 RELATED WORK

**Reward Design.**   The ability of RL agents highly depends on the designs of reward functions. It is widely observed that reward shaping can accelerate learning (Mataric, 1994; Ng et al., 1999; Devlin et al., 2011; Wu & Tian, 2017; Song et al., 2019). Many previous works study how to automatically design auxiliary reward functions for efficient reinforcement learning. A famous paradigm, inverse RL (Ng & Russell, 2000; Fu et al., 2018), considers to recover a reward function from expert demonstrations. Another branch of work is learning an intrinsic reward function that guides the agent to maximize extrinsic objective (Sorg et al., 2010; Guo et al., 2016). Such an intrinsic reward function can be learned through meta-gradients (Zheng et al., 2018; 2020) or self-imitation (Guo et al., 2018; Gangwani et al., 2019). A recent work (Abel et al., 2021) studies the expressivity of Markov rewards and proposes algorithms to design Markov rewards for three notions of abstract tasks.

**Temporal Credit Assignment.**   Another methodology for tackling long-horizon sequential decision problems is assigning credits to emphasize the contribution of each single step over the temporal structure. These methods directly consider the specification of the step values instead of manipulating the reward function. The simplest example is studying how the choice of discount factor $\gamma$ affects the policy learning (Petrik & Scherrer, 2008; Jiang et al., 2015; Fedus et al., 2019). Several previous works consider to extend the $\lambda$-return mechanism (Sutton, 1988) to a more generalized credit assignment framework, such as adaptive $\lambda$ (Xu et al., 2018) and pairwise weights (Zheng et al., 2021). RUDDER (Arjona-Medina et al., 2019) proposes a return-equivalent formulation for the credit assignment problem and establish theoretical analyses (Holzleitner et al., 2021). Aligned-RUDDER (Patil et al., 2020) considers to use expert demonstrations for higher sample efficiency. Harutyunyan et al. (2019) opens up a new family of algorithms, called hindsight credit assignment, that attributes the credits from a backward view. In Appendix F, we cover more topics of related work and discuss the connections to the problem focused by this paper.

## 6 CONCLUSION

In this paper, we propose randomized return decomposition (RRD), a novel reward redistribution algorithm, to tackle the episodic reinforcement learning problem with trajectory feedback. RRD uses a Monte-Carlo estimator to establish a surrogate optimization problem of return decomposition. This surrogate objective implicitly conducts a variance reduction penalty as regularization. We analyze its algorithmic properties by connecting with previous studies in reward redistribution. Our experiments demonstrate that RRD outperforms previous methods in terms of both sample efficiency and policy quality. The basic idea of randomized return decomposition can potentially generalize to other related problems with sum-decomposition structure, such as preference-based reward modeling (Christiano et al., 2017) and multi-agent value decomposition (Sunehag et al., 2018). It is also promising to consider non-linear decomposition as what is explored in multi-agent value factorization (Rashid et al., 2018). We leave these investigations as our future work.

ACKNOWLEDGMENTS

The authors would like to thank Kefan Dong for insightful discussions. This work is supported by the National Science Foundation under Grant CCF-2006526.

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

# A OMITTED PROOFS

**Theorem 1** (Loss Decomposition). *The surrogate loss function $\mathcal{L}_{Rand\text{-}RD}(\theta)$ can be decomposed to two terms interpolating between return decomposition and uniform reward redistribution.*

$$\mathcal{L}_{Rand\text{-}RD}(\theta) = \mathcal{L}_{RD}(\theta) + \underset{\tau \sim \mathcal{D}}{\mathbb{E}} \Bigg[ \underbrace{\underset{\mathcal{I} \sim \rho_T(\cdot)}{Var} \Bigg[ \frac{T}{|\mathcal{I}|} \sum_{t \in \mathcal{I}} \widehat{R}_\theta(s_t, a_t) \Bigg]}_{\text{variance of the Monte-Carlo estimator}} \Bigg] \tag{9}$$

$$= \underbrace{\mathcal{L}_{RD}(\theta)}_{\text{return decomposition}} + \underset{\tau \sim \mathcal{D}}{\mathbb{E}} \Bigg[ T^2 \cdot \underbrace{\underset{(s_t, a_t) \sim \tau}{Var} \Big[ \widehat{R}_\theta(s_t, a_t) \Big]}_{\text{uniform reward redistribution}} \cdot \underbrace{\frac{1}{K} \left( 1 - \frac{K-1}{T-1} \right)}_{\text{interpolation weight}} \Bigg], \tag{10}$$

*where $K$ denotes the length of sampled subsequences defined in Eq. (7).*

*Proof.* First, we note that random sampling serves an unbiased estimation. i.e.,

$$\underset{\mathcal{I} \sim \rho_T(\cdot)}{\mathbb{E}} \Bigg[ \frac{T}{|\mathcal{I}|} \sum_{t \in \mathcal{I}} \widehat{R}_\theta(s_t, a_t) \Bigg] = \sum_{t=0}^{T-1} \widehat{R}_\theta(s_t, a_t) = \widehat{R}_{\text{ep},\theta}(\tau).$$

We can decompose our loss function $\mathcal{L}_{\text{Rand-RD}}(\theta)$ as follows:

$$\mathcal{L}_{\text{Rand-RD}}(\theta) = \underset{\tau \sim \mathcal{D}}{\mathbb{E}} \Bigg[ \underset{\mathcal{I} \sim \rho_T(\cdot)}{\mathbb{E}} \Bigg[ \Bigg( R_{\text{ep}}(\tau) - \frac{T}{|\mathcal{I}|} \sum_{t \in \mathcal{I}} \widehat{R}_\theta(s_t, a_t) \Bigg)^2 \Bigg] \Bigg]$$

$$= \underset{\tau \sim \mathcal{D}}{\mathbb{E}} \Bigg[ \underset{\mathcal{I} \sim \rho_T(\cdot)}{\mathbb{E}} \Bigg[ \Bigg( R_{\text{ep}}(\tau) - \widehat{R}_{\text{ep},\theta}(\tau) + \widehat{R}_{\text{ep},\theta}(\tau) - \frac{T}{|\mathcal{I}|} \sum_{t \in \mathcal{I}} \widehat{R}_\theta(s_t, a_t) \Bigg)^2 \Bigg] \Bigg]$$

$$= \underbrace{\underset{\tau \sim \mathcal{D}}{\mathbb{E}} \Bigg[ \underset{\mathcal{I} \sim \rho_T(\cdot)}{\mathbb{E}} \Bigg[ \Bigg( R_{\text{ep}}(\tau) - \widehat{R}_{\text{ep},\theta}(\tau) \Bigg)^2 \Bigg] \Bigg]}_{=\mathcal{L}_{\text{RD}}(\theta)}$$

$$+ \underset{\tau \sim \mathcal{D}}{\mathbb{E}} \Bigg[ 2 \big( R_{\text{ep}}(\tau) - \widehat{R}_{\text{ep},\theta}(\tau) \big) \cdot \underbrace{\underset{\mathcal{I} \sim \rho_T(\cdot)}{\mathbb{E}} \Bigg[ \Bigg( \widehat{R}_{\text{ep},\theta}(\tau) - \frac{T}{|\mathcal{I}|} \sum_{t \in \mathcal{I}} \widehat{R}_\theta(s_t, a_t) \Bigg) \Bigg]}_{=0} \Bigg]$$

$$+ \underset{\tau \sim \mathcal{D}}{\mathbb{E}} \Bigg[ \underbrace{\underset{\mathcal{I} \sim \rho_T(\cdot)}{\mathbb{E}} \Bigg[ \Bigg( \widehat{R}_{\text{ep},\theta}(\tau) - \frac{T}{|\mathcal{I}|} \sum_{t \in \mathcal{I}} \widehat{R}_\theta(s_t, a_t) \Bigg)^2 \Bigg]}_{=\text{Var}\big[ (T/|\mathcal{I}|) \cdot \sum_{t \in \mathcal{I}} \widehat{R}_\theta(s_t, a_t) \big]} \Bigg]$$

$$= \mathcal{L}_{\text{RD}}(\theta) + \underset{\tau \sim \mathcal{D}}{\mathbb{E}} \Bigg[ \underset{\mathcal{I} \sim \rho_T(\cdot)}{\text{Var}} \Bigg[ \frac{T}{|\mathcal{I}|} \sum_{t \in \mathcal{I}} \widehat{R}_\theta(s_t, a_t) \Bigg] \Bigg].$$

Note our sampling distribution defined in Eq. (7) refers to "*sampling without replacement*" (Singh, 2003) whose variance can be further decomposed as follows:

$$\mathcal{L}_{\text{Rand-RD}}(\theta) = \mathcal{L}_{\text{RD}}(\theta) + \underset{\tau \sim \mathcal{D}}{\mathbb{E}} \Bigg[ \underset{\mathcal{I} \sim \rho_T(\cdot)}{\text{Var}} \Bigg[ \frac{T}{|\mathcal{I}|} \sum_{t \in \mathcal{I}} \widehat{R}_\theta(s_t, a_t) \Bigg] \Bigg]$$

$$= \mathcal{L}_{\text{RD}}(\theta) + \underset{\tau \sim \mathcal{D}}{\mathbb{E}} \Bigg[ T^2 \cdot \underset{(s_t, a_t) \sim \tau}{\text{Var}} \Big[ \widehat{R}_\theta(s_t, a_t) \Big] \cdot \frac{1}{K} \left( 1 - \frac{K-1}{T-1} \right) \Bigg].$$

□

The proof of Theorem 1 follows a particular form of bias-variance decomposition formula (Kohavi & Wolpert, 1996). Similar decomposition form can also be found in other works in the literature of reinforcement learning (Antos et al., 2008).

**Proposition 1** (Surrogate Upper Bound). *The randomized return decomposition loss $\mathcal{L}_{Rand\text{-}RD}(\theta)$ is an upper bound of the actual return decomposition loss function $\mathcal{L}_{RD}(\theta)$, i.e., $\mathcal{L}_{Rand\text{-}RD}(\theta) \geq \mathcal{L}_{RD}(\theta)$.*

*Proof.* Note that the second term of Eq. (9) in Theorem 1 expresses the variance of a Monte-Carlo estimator which is clearly non-negative. It directly gives $\mathcal{L}_{\text{Rand-RD}}(\theta) \geq \mathcal{L}_{\text{RD}}(\theta)$. $\qquad\square$

An alternative proof of Proposition 1 can be directly given by Jensen's inequality.

**Proposition 2** (Objective Gap). *Let $\mathcal{L}_{Rand\text{-}RD}^{(K)}(\theta)$ denote the randomized return decomposition loss that samples subsequences with length $K$. The gap between $\mathcal{L}_{Rand\text{-}RD}^{(K)}(\theta)$ and $\mathcal{L}_{RD}(\theta)$ can be reduced by using larger values of hyper-parameter $K$.*

$$\forall \theta, \quad \mathcal{L}_{Rand\text{-}RD}^{(1)}(\theta) \geq \mathcal{L}_{Rand\text{-}RD}^{(2)}(\theta) \geq \cdots \geq \mathcal{L}_{Rand\text{-}RD}^{(T-1)}(\theta) \geq \mathcal{L}_{Rand\text{-}RD}^{(T)}(\theta) = \mathcal{L}_{RD}(\theta). \tag{11}$$

*Proof.* In Eq. (10) of Theorem 1, the last term $\frac{1}{K}\left(1 - \frac{K-1}{T-1}\right)$ monotonically decreases as the hyper-parameter $K$ increases. When $K = T$, this coefficient is equal to zero. It derives Eq. (11) in the given statement. $\qquad\square$

**Proposition 3** (Uniform Reward Redistribution). *Assume all trajectories have the same length and the parameterization space of $\theta$ serves universal representation capacity. The optimal solution $\theta^\star$ of minimizing $\mathcal{L}_{Rand\text{-}RD}^{(1)}(\theta)$ is stated as follows:*

$$\widehat{R}_{\theta^\star}(s,a) = \mathop{\mathbb{E}}_{\tau \sim \mathcal{D}}\big[R_{ep}(\tau)/T \mid (s,a) \in \tau\big], \tag{12}$$

*where $\mathcal{L}_{Rand\text{-}RD}^{(1)}(\theta)$ denotes the randomized return decomposition loss with $K = 1$.*

*Proof.* Note that we assume all trajectories have the same length. The optimal solution of this least-squares problem is given by

$$\begin{aligned}
\widehat{R}_{\theta^\star}(s,a) &= \min_{r \in \mathbb{R}} \mathop{\mathbb{E}}_{\tau \sim \mathcal{D}} \left[ (R_{\text{ep}}(\tau) - T \cdot r)^2 \,\big|\, (s,a) \in \tau \right] \\
&= \min_{r \in \mathbb{R}} \mathop{\mathbb{E}}_{\tau \sim \mathcal{D}} \left[ \frac{1}{T^2}\,(R_{\text{ep}}(\tau)/T - r)^2 \,\big|\, (s,a) \in \tau \right] \\
&= \min_{r \in \mathbb{R}} \frac{1}{T^2} \mathop{\mathbb{E}}_{\tau \sim \mathcal{D}} \left[ (R_{\text{ep}}(\tau)/T - r)^2 \,\big|\, (s,a) \in \tau \right] \\
&= \min_{r \in \mathbb{R}} \mathop{\mathbb{E}}_{\tau \sim \mathcal{D}} \left[ (R_{\text{ep}}(\tau)/T - r)^2 \,\big|\, (s,a) \in \tau \right] \\
&= \mathop{\mathbb{E}}_{\tau \sim \mathcal{D}} \left[ R_{\text{ep}}(\tau)/T \mid (s,a) \in \tau \right],
\end{aligned}$$

which depends on the trajectory distribution in dataset $\mathcal{D}$. $\qquad\square$

If we relax the assumption that all trajectories have the same length, the solution of the above least-squares problem would be a weighted expectation as follows:

$$\begin{aligned}
\widehat{R}_{\theta^\star}(s,a) &= \min_{r \in \mathbb{R}} \mathop{\mathbb{E}}_{\tau \sim \mathcal{D}} \left[ (R_{\text{ep}}(\tau) - T_\tau \cdot r)^2 \,\big|\, (s,a) \in \tau \right] \\
&= \min_{r \in \mathbb{R}} \mathop{\mathbb{E}}_{\tau \sim \mathcal{D}} \left[ T_\tau^2 \cdot (R_{\text{ep}}(\tau)/T_\tau - r)^2 \,\big|\, (s,a) \in \tau \right] \\
&= \frac{\sum_{\tau \in \mathcal{D}:(s,a) \in \tau} T_\tau \cdot R_{\text{ep}}(\tau)}{\sum_{\tau \in \mathcal{D}:(s,a) \in \tau} T_\tau^2},
\end{aligned}$$

where $T_\tau$ denotes the length of trajectory $\tau$. This solution can still be interpreted as a uniform reward redistribution, in which the dataset distribution is prioritized by the trajectory length.

## B  EXPERIMENT SETTINGS AND IMPLEMENTATION DETAILS

### B.1  MUJOCO BENCHMARK WITH EPISODIC REWARDS

**MuJoCo Benchmark with Episodic Rewards.**   We adopt the same experiment setting as Gangwani et al. (2020) and compare our approach with baselines in a suite of MuJoCo locomotion benchmark tasks with episodic rewards. This experiment setting is commonly used in the literature (Mania et al., 2018; Guo et al., 2018; Liu et al., 2019; Arjona-Medina et al., 2019; Gangwani et al., 2019; 2020). The environment simulator is based on OpenAI Gym (Brockman et al., 2016). These tasks are long-horizon with maximum trajectory length $T = 1000$. We modify the reward function of these environments to set up an episodic-reward setting. Formally, on non-terminal states, the agent will receive a zero signal instead of the per-step dense rewards. The agent can obtain the episodic feedback $R_{\mathrm{ep}}(\tau)$ at the last step of the rollout trajectory, in which $R_{\mathrm{ep}}(\tau)$ is computed by the summation of per-step instant rewards given by the standard setting.

**Hyper-Parameter Configuration For MuJoCo Experiments.**   In MuJoCo experiments, the policy optimization module of RRD is implemented based on soft actor-critic (SAC; Haarnoja et al., 2018a). We evaluate the performance of our proposed methods with the same configuration of hyper-parameters in all environments. The hyper-parameters of the back-end SAC follow the official technical report (Haarnoja et al., 2018b). We summarize our default configuration of hyper-parameters as the following table:

| Hyper-Parameter | Default Configuration |
|---|---|
| discount factor $\gamma$ | 0.99 |
| # hidden layers (all networks) | 2 |
| # neurons per layer | 256 |
| activation | ReLU |
| optimizer (all losses) | Adam (Kingma & Ba, 2015) |
| learning rate | $3 \cdot 10^{-4}$ |
| initial temperature $\alpha_{\mathrm{init}}$ | 1.0 |
| target entropy | $-\dim(\mathcal{A})$ |
| Polyak-averaging coefficient | 0.005 |
| # gradient steps per environment step | 1 |
| # gradient steps per target update | 1 |
| # transitions in replay buffer | $10^6$ |
| # transitions in mini-batch for training SAC | 256 |
| # transitions in mini-batch for training $\widehat{R}_\theta$ | 256 |
| # transitions per subsequence ($K$) | 64 |
| # subsequences in mini-batch for training $\widehat{R}_\theta$ | 4 |

Table 1: The hyper-parameter configuration of RRD in MuJoCo experiments.

In addition to SAC, we also provide the implementations upon DDPG (Lillicrap et al., 2016) and TD3 (Fujimoto et al., 2018) in our Github repository.

### B.2  ATARI BENCHMARK WITH EPISODIC REWARDS

**Atari Benchmark with Episodic Rewards.**   In addition, we conduct experiments in a suite of Atari games with episodic rewards. The environment simulator is based on OpenAI Gym (Brockman et al., 2016). Following the standard Atari pre-processing proposed by Mnih et al. (2015), we rescale each RGB frame to an $84 \times 84$ luminance map, and the observation is constructed as a stack of 4 recent luminance maps. We modify the reward function of these environments to set up an episodic-reward setting. Formally, on non-terminal states, the agent will receive a zero signal instead of the per-step dense rewards. The agent can obtain the episodic feedback $R_{\mathrm{ep}}(\tau)$ at the last step of the rollout trajectory, in which $R_{\mathrm{ep}}(\tau)$ is computed by the summation of per-step instant rewards given by the standard setting.

**Hyper-Parameter Configuration For Atari Experiments.** In Atari experiments, the policy optimization module of RRD is implemented based on deep Q-network (DQN; Mnih et al., 2015). We evaluate the performance of our proposed methods with the same configuration of hyper-parameters in all environments. The hyper-parameters of the back-end DQN follow the technical report (Castro et al., 2018). We summarize our default configuration of hyper-parameters as the following table:

| Hyper-Parameter | Default Configuration |
|---|---|
| discount factor $\gamma$ | 0.99 |
| # stacked frames in agent observation | 4 |
| # `noop` actions while starting a new episode | 30 |
| network architecture | DQN (Mnih et al., 2015) |
| optimizer for Q-values | Adam (Kingma & Ba, 2015) |
| learning rate for Q-values | $6.25 \cdot 10^{-5}$ |
| optimizer for $\widehat{R}_\theta$ | Adam (Kingma & Ba, 2015) |
| learning rate for $\widehat{R}_\theta$ | $3 \cdot 10^{-4}$ |
| exploration strategy | $\epsilon$-greedy |
| $\epsilon$ decaying range - start value | 1.0 |
| $\epsilon$ decaying range - end value | 0.01 |
| # timesteps for $\epsilon$ decaying schedule | 250000 |
| # gradient steps per environment step | 0.25 |
| # gradient steps per target update | 8000 |
| # transitions in replay buffer | $10^6$ |
| # transitions in mini-batch for training DQN | 32 |
| # transitions in mini-batch for training $\widehat{R}_\theta$ | 32 |
| # transitions per subsequence ($K$) | 32 |
| # subsequences in mini-batch for training $\widehat{R}_\theta$ | 1 |

Table 2: The hyper-parameter configuration of RRD in Atari experiments.

### B.3 AN ALTERNATIVE IMPLEMENTATION OF RANDOMIZED RETURN DECOMPOSITION

Recall that the major practical barrier of the least-squares-based return decomposition method specified by $\mathcal{L}_{\text{RD}}(\theta)$ is its scalability in terms of the computation costs. The trajectory-wise episodic reward is the only environmental supervision for reward modeling. Computing the loss function $\mathcal{L}_{\text{RD}}(\theta)$ with a single episodic reward label requires to enumerate all state-action pairs along the whole trajectory.

Theorem 1 motivates an unbiased implementation of randomized return decomposition that optimizes $\mathcal{L}_{\text{RD}}(\theta)$ instead of $\mathcal{L}_{\text{Rand-RD}}(\theta)$. By rearranging the terms of Eq. (10), we can obtain the difference between $\mathcal{L}_{\text{Rand-RD}}(\theta)$ and $\mathcal{L}_{\text{RD}}(\theta)$ as follows:

$$\mathcal{L}_{\text{Rand-RD}}(\theta) - \mathcal{L}_{\text{RD}}(\theta) = \underset{\mathcal{I} \sim \rho_T(\cdot)}{\text{Var}} \left[ \frac{T}{|\mathcal{I}|} \sum_{t \in \mathcal{I}} \widehat{R}_\theta(s_t, a_t) \right].$$

Note our sampling distribution $\rho_T(\cdot)$ defined in Eq. (7) refers to "*sampling without replacement*" (Singh, 2003) whose variance can be estimated as follows:

$$\underset{\mathcal{I} \sim \rho_T(\cdot)}{\text{Var}} \left[ \frac{T}{|\mathcal{I}|} \sum_{t \in \mathcal{I}} \widehat{R}_\theta(s_t, a_t) \right] = T^2 \cdot \underset{\mathcal{I} \sim \rho_T(\cdot)}{\mathbb{E}} \left[ \frac{T - K}{T} \cdot \frac{\sum_{t \in \mathcal{I}} \left( \widehat{R}_\theta(s_t, a_t) - \bar{R}_\theta(\mathcal{I}; \tau) \right)^2}{K(K - 1)} \right],$$

where $\bar{R}_\theta(\mathcal{I}; \tau) = \frac{1}{|\mathcal{I}|} \sum_{t \in \mathcal{I}} \widehat{R}_\theta(s_t, a_t)$. Thus we can obtain an unbiased estimation of this variance penalty by sampling a subsequence $\mathcal{I}$. By subtracting this estimation from $\widehat{\mathcal{L}}_{\text{Rand-RD}}(\theta)$, we can

obtain an unbiased estimation of $\mathcal{L}_{\text{RD}}(\theta)$. More specifically, we can use the following sample-based loss function to substitute Eq. (13) in implementation:

$$\widehat{\mathcal{L}}_{\text{RD}}(\theta) = \underbrace{\frac{1}{M}\sum_{j=1}^{M}\left[\left(R_{\text{ep}}(\tau_j) - \frac{T_j}{|\mathcal{I}_j|}\sum_{t\in\mathcal{I}_j}\widehat{R}_\theta(s_{j,t},a_{j,t})\right)^2\right]}_{\widehat{\mathcal{L}}_{\text{Rand-RD}}(\theta)}$$
$$- \frac{1}{M}\sum_{j=1}^{M}\left[\frac{T_j(T_j-K)}{|\mathcal{I}_j|(|\mathcal{I}_j|-1)}\cdot\sum_{t\in\mathcal{I}_j}\left(\widehat{R}_\theta(s_{j,t},a_{j,t}) - \frac{1}{|\mathcal{I}_j|}\sum_{t\in\mathcal{I}_j}\widehat{R}_\theta(s_{j,t},a_{j,t})\right)^2\right].$$

The above loss function can be optimized through the same mini-batch training paradigm as what is presented in Algorithm 1.

## C   EXPERIMENTS ON ATARI BENCHMARK WITH EPISODIC REWARDS

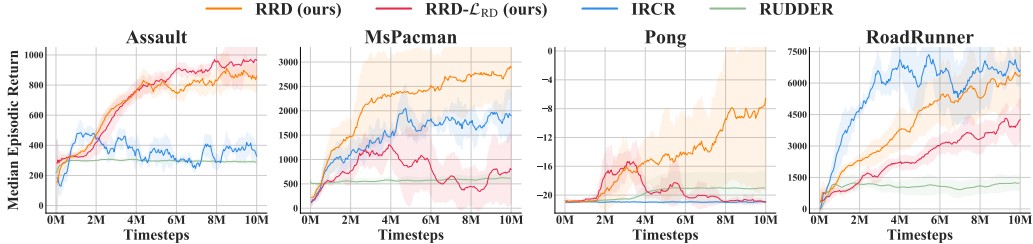

Figure 3: Learning curves on a suite of Atari benchmark tasks with episodic rewards. These curves are plotted from 5 runs with random initializations. The shaded region indicates the standard deviation. To make the comparison more clear, the curves are smoothed by averaging 10 most recent evaluation points. We set up an evaluation point every $5 \cdot 10^4$ timesteps.

Note that our proposed method does not restricts its usage to continuous control problems. It can also be integrated in DQN-based algorithms to solve problems with discrete-action space. We evaluate the performance of our method built upon DQN in several famous Atari games. The reward redistribution problem in these tasks is more challenging than that in MuJoCo locomotion benchmark since the task horizon of Atari is much longer. For example, the maximum task horizon in Pong can exceed 20000 steps in a single trajectory. This setting highlights the scalability advantage of our method, i.e., the objective of RRD can be optimized by sampling short subsequences whose computation cost is manageable. The experiment results are presented in Figure 3. Our method outperforms all baselines in 3 out of 4 tasks. We note that IRCR outperforms RRD in RoadRunner. It may be because IRCR is non-parametric and thus does not suffer from the difficulty of processing visual observations.

## D   VISUALIZING THE PROXY REWARDS OF RRD

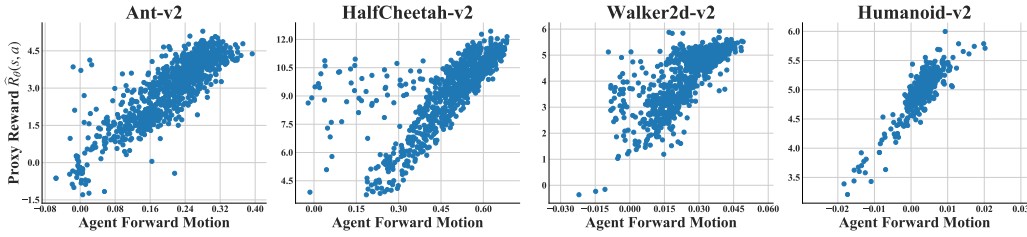

Figure 4: Visualization of the proxy rewards learned by RRD in MuJoCo locomotion tasks.

In MuJoCo locomotion tasks, the goal of agents is running towards a fixed direction. In Figure 4, we visualize the correlation between per-step forward distance and the assigned proxy reward. We uniformly collected $10^3$ samples during the first $10^6$ training steps. "Agent Forward Motion" denotes the forward distance at a single step. "Proxy Reward $\widehat{R}_\theta(s, a)$" denotes the immediate proxy reward assigned at that step. It shows that the learned proxy reward has high correlation to the forward distance at that step.

# E    AN ABLATION STUDY ON THE HYPER-PARAMETERS OF IRCR

We note that Gangwani et al. (2020) uses a different hyper-parameter configuration from the standard SAC implementation (Haarnoja et al., 2018b). The differences exist in two hyper-parameters:

| Hyper-Parameter | Default Configuration |
| --- | --- |
| Polyak-averaging coefficient | 0.001 |
| # transitions in replay buffer | $3 \cdot 10^5$ |
| # transitions in mini-batch for training SAC | 512 |

Table 3: The hyper-parameters used by Gangwani et al. (2020) in MuJoCo experiments.

To establish a rigorous comparison, we evaluate the performance of IRCR with the hyper-parameter configuration proposed by Haarnoja et al. (2018b), so that IRCR and RRD use the same hyper-parameters in their back-end SAC agents. The experiment results are presented in Figure 5.

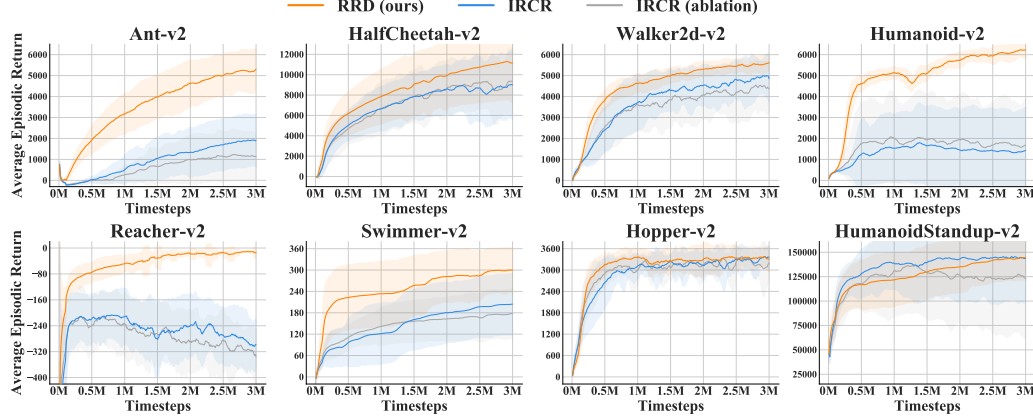

Figure 5: An ablation study on the hyper-parameter configuration of IRCR. The curves of "IRCR" refer to the performance of IRCR using the hyper-parameter setting proposed by Gangwani et al. (2020). The curves of "IRCR (ablation)" refer to the performance of IRCR using the hyper-parameters stated in Table 1. All curves are plotted from 30 runs with random initializations.

As shown in Figure 5, the hyper-parameters tuned by Gangwani et al. (2020) is more stable in most environments. Although using hyper-parameters stated in Table 1 can improve the performance in some cases, the overall performance cannot outperform RRD.

# F    RELATED WORK

**Delayed Feedback.**    Tackling environmental delays is a long-lasting problem in reinforcement learning and control theory (Nilsson et al., 1998; Walsh et al., 2009; Zhou et al., 2018; 2019a; Héliou et al., 2020; Nath et al., 2021; Tang et al., 2021). In real-world applications, almost all environmental signals have random delays (Schuitema et al., 2010; Hester & Stone, 2013; Amuru & Buehrer, 2014; Lei et al., 2020), which is a fundamental challenge for the designs of RL algorithms. A classical method to handle delayed signals is stacking recent observations within a small sliding window as

the input for decision-making (Katsikopoulos & Engelbrecht, 2003). This simple transformation can establish a Markovian environment formulation, which is widely used to deal with short-term environmental delays (Mnih et al., 2015). Many recent works focus on establishing sample-efficient off-policy RL algorithm that is adaptive to delayed environmental signals (Bouteiller et al., 2021; Han et al., 2021). In this paper, we consider an extreme delay of reward signals, which is a harder problem setting than short-term random delays.

**Reward Design.**    The ability of reinforcement learning agents highly depends on the designs of reward functions. It is widely observed that reward shaping can accelerate learning (Mataric, 1994; Ng et al., 1999; Devlin et al., 2011; Wu & Tian, 2017; Song et al., 2019). Many previous works study how to automatically design auxiliary reward functions for efficient reinforcement learning. A famous paradigm, inverse reinforcement learning (Ng & Russell, 2000; Fu et al., 2018), considers to recover a reward function from expert demonstrations. Several works consider to learn a reward function from expert labels of preference comparisons (Wirth et al., 2016; Christiano et al., 2017; Lee et al., 2021), which is a form of weak supervision. Another branch of work is learning an intrinsic reward function from experience that guides the agent to maximize extrinsic objective (Sorg et al., 2010; Guo et al., 2016). Such an intrinsic reward function can be learned through meta-gradients (Zheng et al., 2018; 2020) or self-imitation (Guo et al., 2018; Gangwani et al., 2019). A recent work (Abel et al., 2021) studies the expressivity of Markov rewards and proposes algorithms to design Markov rewards for three notions of abstract tasks.

**Temporal Credit Assignment.**    Another methodology for tackling long-horizon sequential decision problems is assigning credits to emphasize the contribution of each single step over the temporal structure. These methods directly consider the specification of the step values instead of manipulating the reward function. The simplest example is studying how the choice of discount factor $\gamma$ affects the policy learning (Petrik & Scherrer, 2008; Jiang et al., 2015; Fedus et al., 2019). Several previous works consider to extend the $\lambda$-return mechanism (Sutton, 1988) to a more generalized credit assignment framework, such as adaptive $\lambda$ (Xu et al., 2018) and pairwise weights (Zheng et al., 2021). RUDDER (Arjona-Medina et al., 2019) proposes a return-equivalent formulation for the credit assignment problem and establish theoretical analyses (Holzleitner et al., 2021). Aligned-RUDDER (Patil et al., 2020) considers to use expert demonstrations for higher sample efficiency. Harutyunyan et al. (2019) opens up a new family of algorithms, called hindsight credit assignment, that attributes the credits from a backward view.

**Value Decomposition.**    This paper follows the paradigm of reward redistribution that aims to decompose the return value to step-wise reward signals. The simplest mechanism in the literature is the uniform reward redistribution considered by Gangwani et al. (2020). It can be effectively integrated with off-policy reinforcement learning and thus achieves state-of-the-art performance in practice. Least-squares-based reward redistribution is investigated by Efroni et al. (2021) from a theoretical point of view. Chatterji et al. (2021) extends the theoretic results to the logistic reward model. In game theory and multi-agent reinforcement learning, a related problem is how to attribute a global team reward to individual rewards (Nguyen et al., 2018; Du et al., 2019; Wang et al., 2020), which provide agents incentives to optimize the global social welfare (Vickrey, 1961; Clarke, 1971; Groves, 1973; Myerson, 1981). A promising paradigm for multi-agent credit assignment is using structural value representation (Sunehag et al., 2018; Rashid et al., 2018; Son et al., 2019; Böhmer et al., 2020; Wang et al., 2021a;b), which supports end-to-end temporal difference learning. This paradigm transforms the value decomposition to the structured prediction problem. A future work is integrating prior knowledge of the decomposition structure as many previous works for structured prediction (Chen et al., 2020; Tavakoli et al., 2021).

