# OpenReview forum: "Learning Long-Term Reward Redistribution via Randomized Return Decomposition"
_ICLR.cc/2022/Conference — ICLR 2022 Spotlight_

### Official Review · Reviewer_73tN · 2021-11-02

**Correctness:** 2
**Technical Novelty And Significance:** 2
**Empirical Novelty And Significance:** 2
**Recommendation:** 8
**Confidence:** 3

**Main Review:**

Minor nitpick: The theta used in (4) should be formally defined.  One potential confusion that could result from not defining it: theta could parameterize the policy that generates the trajectory tau, or it could parameterize \hat R (I know it’s the latter, but I think this is a potential point of confusion that can be avoided by defining things more formally).

The notation around (6) is a little confusing, \mathcal I is not defined formally (it is a subset of \mathcal Z_T, but that’s not defined) or intuitively in English, except to say that it is sampled from rho_T.  rho_T “denotes an unbiased sampling distribution”, but of what?  I can infer that it is of the timesteps in the trajectory, (and thus I can infer the meaning \mathcal I) but all of this should be made clearer; it’s best not to make the reader have to guess and infer definitions.

Minor grammar issue: “Despite the Monte-Carlo estimator is unbiased,”

Questions:
- The RRD-L_RD is claimed as the authors’ algorithm, but that loss function is discussed as prior work when it is given in (4).  What is the contribution that makes it algorithm authors’ (and not prior work)?  The answer to this question might be a nice addition to supplementary material section B.2.
- “We evaluate the performance of our proposed methods with the same configuration of
hyper-parameters in all environments.”  To clarify, all algorithms in the experiments (not just your two proposed methods) used these hyperparameters, correct?
- Was any hyperparameter tuning performed?  How were the hyperparameters chosen?
- How many runs were used in each experiment? (I know at least 10, but more specifics would be good.)  What criteria were used for choosing to use different numbers of runs for different experiments?  I am concerned there could be bias here: 10 runs is not a large number for RL, where variance between runs is often enormous, and if some plots or curves were given additional runs after the authors viewed the results for 10 runs, that could add significant bias to the results in favor of their methods.
- Why the unusual choice for the error bars?  I suspect the standard choice (standard deviation) did not tell the desired story as convincingly…  (Or is this a common choice that I have simply not encountered much?)

**Update after the rebuttal and reading the other reviews:**

Most of my concerns have been addressed.  However, the fact that most of the plots used 10 runs raises a new concern: statistical significance.  (I did not raise this point before, "at least 10" could have meant 30+ for most plots.)  RL experiments are known to have large variance between runs, often with rare but extreme "outlier runs", so the common practice of including only 1-10 runs is not good, as has been pointed out in many papers, talks, etc. in recent years.  30+ runs (and ideally some statistical significance testing) would make this paper much stronger.  For this reason, I am only raising my score to weak accept (6) instead of accept (8).  (I would maintain my score of 5, but the paper is strong in other areas, and 10 runs is not quite as bad as many papers which include only 1-5 runs, and the trends that the authors assert seem somewhat clear across environments.)

**Update after final comments below:**

I am updating my score to an 8; see below.

**Summary Of The Paper:**

The authors propose a method of constructing a proxy reward function (and the loss function for learning that reward function), which generalizes two prior methods, allowing it to trade off the advantages and disadvantages of those methods.  They create an RL algorithm based on this, analyze it theoretically and relative to prior work, and show that it performs well empirically.


**Summary Of The Review:**

The overall contribution is a little light, but this is nonetheless a solid well-executed paper as a whole.  However, due to the questions above about experiments, I cannot accept the paper in its current form.  Pending the answers to those questions (and the reviewer discussion), I may raise my score significantly.

---

> ### Author Response · Authors · 2021-11-23
> **Response to Reviewer 73tN**
>
> Thanks for the inspiring comments. We provide clarification to your questions and concerns as below. If our response does not fully address your concerns, please post additional questions and we will be happy to have further discussions.
>
> **Q1: Notation issues**
>
> We thank the reviewer for pointing out these writing flaws. The reviewer's understanding of notations is correct. We have revised the notation definition around Eq.(4) and Eq.(6).
>
> **Q2: The RRD-$\mathcal{L}_{RD}$ is claimed as the authors’ algorithm, but that loss function is discussed as prior work when it is given in (4).**
>
> The loss function $\mathcal{L}_{RD}$ is introduced in prior work but this loss function cannot be computed efficiently when the agent trajectory is very long. More specifically, the computation cost for utilizing each episodic reward label is proportional to the trajectory length $T$.
>
> RRD-$\mathcal{L}\_{\text{RD}}$ is a computationally efficient method to optimize $\mathcal{L}\_{RD}$. It runs in the same nature of RRD-$\mathcal{L}\_{\text{Rand-RD}}$ using randomized return decomposition, which estimates loss function by sampling short subsequences. The difference of RRD-$\mathcal{L}\_{\text{RD}}$ from RRD-$\mathcal{L}\_{\text{Rand-RD}}$ is an extra loss term that estimates the objective gap in Eq.(9), i.e., the closed-form formula of $(\mathcal{L}\_{\text{RD}}-\mathcal{L}\_{\text{Rand-RD}})$ give by Theorem 1.
>
> In this revision, we enrich the description about RRD-$\mathcal{L}_{\text{RD}}$ in Appendix B.2 and refine the discussion in section 4.1.
>
> **Q3\&4: Clarifications on hyper-parameters**
>
> In the implementation of RRD, the configuration of hyper-parameters can be roughly divided into three folds:
> - The hyper-parameters of SAC follow the official technical report [1]. **We did not perform any hyper-parameter tuning for the SAC agent.**
> - The hyper-parameters of reward model $\widehat R_\theta$. The network architecture of the reward model is the same as the critic network of SAC. The training procedure of the reward model is also the same as that of critic values, i.e.,  using the same optimizer, the same learning rate, the same replay buffer, the same batch size, etc. **We did not perform any hyper-parameter tuning for the reward model.**
> - The only special hyper-parameter of RRD is the length of subsequences (denoted by $K$). We include an ablation study in section 4.2. It shows that this hyper-parameter is not sensitive in a reasonable range.
>
> The hyper-parameters of baseline algorithms are determined as follows:
> - IRCR is an off-policy approach built upon SAC, which is the most closely related method to RRD. The presented curves of IRCR use the hyper-parameters tuned by IRCR's authors. In Appendix E, we show that IRCR would perform worse when using the standard hyper-parameter configuration of SAC [1].
> - Other three baselines are all on-policy algorithms, which are built upon PPO. Since they do not follow the same training paradigm as RRD, we just follow their original implementations to evaluate their performance.
>
> **Q5: Clarifications on the number of runs**
>
> In the initial submission, we include 20 runs for IRCR in \{*Walker2d-v2, Swimmer-v2, Humanoid-v2, HumanoidStandup-v2*\} due to the large variance. All other curves contain exactly 10 runs.
>
> In this revision, we remove those additional runs of IRCR so that all curves are plotted by 10 runs. More specifically, for those IRCR curves with 20 runs initially, we plot the evaluation using the first collected 10 runs and delete the other data.
>
> Since the performance gaps between our method and baselines are quite significant, **this modification does not alter the conclusion of experiments.**

---

> > ### Author Response · Authors · 2021-11-23
> > **Response to Reviewer 73tN**
> >
> > **Q6: Clarifications on the plot style**
> >
> > In this revision, we replace all curves by using the mean-std metric. **This modification on plot style does not alter the conclusion of experiments.**
> >
> > Since the reviewer suspects our motivation of using median-based metric, we would like to argue a little bit on the advantages of evaluating median performance.
> > - First, the standard deviation metric implicitly assumes that the collected samples follow a normal distribution, which is not realistic in practice. More specifically, the empirical computation of mean and standard deviation tries to model the collected samples by a Gaussian random variable.
> > - In our initial submission, we plot the median performance and 60\% population around the median. Such a ranged-based metric is more rigorous when the data do not follow normal distributions. It directly shows the center of collected samples.
> >
> > Please refer to this paper [2] published in *ICLR Workshop on Reproducibility* for more discussions. Since both metrics are acceptable from the statistical point of view, we have modified the plot metric to the mean-std style, so that our presentation style becomes closer to the literature convention.
> >
> > If this revision does not address the reviewer's concerns, please post additional questions and we will be happy to have further discussions.
> >
> > **References**
> >
> > [1] Haarnoja, Tuomas, et al. "Soft actor-critic algorithms and applications." arXiv preprint arXiv:1812.05905 (2018).
> >
> > [2] Colas, Cédric, Olivier Sigaud, and Pierre-Yves Oudeyer. "A Hitchhiker's Guide to Statistical Comparisons of Reinforcement Learning Algorithms." ICLR Workshop on Reproducibility. 2019.

---

> > > ### Comment · Reviewer_73tN · 2021-11-25
> > > **Plots**
> > >
> > > Thank you for your hard work; I have updated my review, see above.  Regarding the plots: your logic is quite reasonable; please feel free to revert them to the earlier form if you think that is best.

---

> > > > ### Author Response · Authors · 2021-11-30
> > > > **Thanks for the insightful comments. We will continue to add more experiment runs.**
> > > >
> > > > Thanks for your time and efforts in reviewing our work. The insightful comments help us to improve our work.
> > > >
> > > > Regarding the post-rebuttal comments, we add additional runs to our main experiments in Figure 1. The following table presents the mean-std performance of RRD and IRCR when completing 3M steps of training, which is evaluated over 30 runs. The conclusion of experiments does not alter.
> > > >
> > > > | Method | Ant-v2 | HalfCheetah-v2 | Walker2d-v2 | Humanoid-v2 |
> > > > | ------ |  ------ |  ------ |  ------ |  ------ |
> > > > | RRD (ours) | $5304\pm 976.0$ | $11157.5\pm 3877.9$ | $5618.2\pm 379.6$ | $6211.0\pm 410.7$ |
> > > > | IRCR (baseline) | $2750.4\pm 1292.6$ | $9001.3\pm 3410.4$ | $4950.0\pm 1103.5$ | $3164.5\pm 2354.6$ |
> > > >
> > > > Since it requires a large demand of computation resources (8 environments $\times$ 6 algorithms $+$ ablation studies), we will continually work on it after the rebuttal period to make sure every curve in our paper contains 30 runs in the next revision.

---

> > > > > ### Comment · Reviewer_73tN · 2021-11-30
> > > > > **Great!**
> > > > >
> > > > > This looks great and is convincing.  Since you have promised to do this for all curves, I have updated my score to an 8.  Thank you for your hard work.

---

### Official Review · Reviewer_UMu2 · 2021-11-02

**Correctness:** 3
**Technical Novelty And Significance:** 2
**Empirical Novelty And Significance:** 3
**Recommendation:** 5
**Confidence:** 4

**Main Review:**

- Experiments:

Experiments are well done and satisfactory. It seems all the environments have continuous action space. It would be interesting to know how the algorithm behaves for discrete action space.

- Credit assignment and reward functions:

	Through the entire paper, credit assignment and reward functions have been used interchangeably. Constructing reward functions and assigning credit are two different problems. Also, RUDDER does not try to create a new dense reward function. It tries to assign credit to actions and it does this via decomposing the return of a trajectory. This has to be corrected in the paper.

- RUDDER markov proxy reward function:

  RUDDER clearly states that the reward redistribution is second order markov (Theorem 2 in Arjona-Medina et.al). While in the paper it is mentioned that the resulting reward is a “Markovian proxy reward function”. Please provide clarification on why this is the case?

- Return Equivalence
 	One of the ways RUDDER ensures that the optimal policy stays the same is by being return equivalent. It is not clear how RRD ensures return equivalence or ensures that the optimal policy stays the same.

- Equation (4)
	The section where equation (4) is introduced seems to imply that all methods use such a loss function. For example, RUDDER does not use this loss and should be clarified.

Missing related work:

[1] also does reward redistribution for credit assignment for the complex task of Minecraft. [2] also looks at the problem of credit assignment and is relevant to your work.

[1] Align-Rudder: Learning from few demonstrations by reward redistribution

[2] Hindsight credit assignment


**Summary Of The Paper:**

The paper addresses the problem of credit assignment in delayed reward setting. It does this by providing a new mechanism for reward redistribution. The authors claim that the new mechanism makes reward redistribution more scalable. The authors predict the return of a trajectory by using only random sub-sequences in the trajectory. Then the prediction model is used for assigning reward for state-action pairs.


**Summary Of The Review:**

Overall, the paper provides a new mechanism to reduce the delay in the reward and make learning faster in delayed settings. But the work is not too novel and the writing is not satisfactory. It is not clear why the optimal policy does not change after such randomized reward redistribution.

---

> ### Author Response · Authors · 2021-11-23
> **Response to Reviewer UMu2**
>
> Thanks for the inspiring comments. We provide clarification to your questions and concerns as below. If our response does not fully address your concerns, please post additional questions and we will be happy to have further discussions.
>
> **Q1: It would be interesting to know how the algorithm behaves for discrete action space.**
>
> In Appendix C, we add experiments on several Atari games. We modify the reward function to an episodic reward setting as what we did in MuJoCo benchmark. These video games are more challenging than MuJoCo locomotion tasks, since (1) they have longer task horizon and (2) the ground-truth per-step rewards are specified by complicated logics.
>
> We evaluate the performance of RRD built upon DQN and compare its performance against baseline algorithms. As shown in Figure 3, our method significantly outperforms IRCR and RUDDER. We will include more experiments on Atari benchmark in the next revision.
>
> **Q2: Through the entire paper, credit assignment and reward functions have been used interchangeably. Constructing reward functions and assigning credit are two different problems. RUDDER does not try to create a new dense reward function. It tries to assign credit to actions and it does this via decomposing the return of a trajectory. This has to be corrected in the paper.**
>
> Thanks for this kind remind. We refine the usage of these terminologies to reduce the ambiguity. In the rebuttal revision, we only mention the terminology of "credit assignment" when discussing this branch of related work. We also pay special attention to the clarification of RUDDER in section 1, 2, and 4. We avoid referring RUDDER when discussing the construction of proxy rewards.
>
> **Q3: RUDDER clearly states that the reward redistribution is second order markov. While in the paper it is mentioned that the resulting reward is a “Markovian proxy reward function”. Please provide clarification on why this is the case?**
>
> From the technical view, this paper follows a different methodology from RUDDER. The purpose of citing RUDDER in the main text is acknowledging the idea of return decomposition for reward redistribution. In the rebuttal revision, we refine the discussions around these terminologies to reduce the ambiguity.
>
> The concept of "Markovian proxy reward function" is pursued by the literature of online reward design [1]. This Markovian-version definition is preferable in practice, since it is simpler to optimize and is more compatible with advanced off-policy RL algorithms such as SAC.
>
> **Q4: RUDDER ensures that the optimal policy stays the same is by being return equivalent. It is not clear how RRD ensures return equivalence or ensures that the optimal policy stays the same.**
>
> We do not claim the optimal policy stays the same while using the learned proxy rewards. None of our theory statements claim or infer this property.
>
> We find this misunderstanding may come from the introduction section of the initial submission. We introduce the equivalence of optimal policy as an intuitive motivation of online reward design. This paradigm designs its objective  In the rebuttal revision, we refine the discussion to remove this ambiguity.
>
> In this paper, we focus on the trade-off between computation costs and reward misspecification. We aim to design a reward redistribution algorithm that can be easily implemented in long-horizon tasks with manageable computation costs. Meanwhile, we hope the redistributed rewards can express the original goal of the given task as well as possible. Our analyses characterize the deviation of our objective $\mathcal{L}\_{\text{Rand-RD}}$ from the original return decomposition loss $\mathcal{L}_{\text{RD}}$ . Our experiments demonstrate that the approximation made by RRD is acceptable in practice.
>
> **Q5: The section where equation (4) is introduced seems to imply that all methods use such a loss function. For example, RUDDER does not use this loss and should be clarified.**
>
> In the rebuttal revision, we refine the discussion around the loss function Eq.(4). We refer this loss function to Efroni et al. [2] instead of the general return decomposition idea. In addition, we refine the discussion in section 4 to clarify that RUDDER does not optimize $\mathcal{L}_{\text{RD}}$.
>
> **Q6: Missing related work**
>
> We thank the reviewer for additional related work. It helps us to improve our presentation. We have included them in section 5 and present an extended related work section in Appendix F.
>
> If this revision does not address the reviewer's concerns, please post additional questions and we will be happy to have further discussions.
>
> **References**
>
> [1] Zheng, Zeyu, Junhyuk Oh, and Satinder Singh. "On Learning Intrinsic Rewards for Policy Gradient Methods." Advances in Neural Information Processing Systems 31 (2018)
>
> [2] Efroni, Yonathan, Nadav Merlis, and Shie Mannor. "Reinforcement Learning with Trajectory Feedback." Proceedings of the AAAI Conference on Artificial Intelligence. 2021.

---

> > ### Comment · Reviewer_UMu2 · 2021-11-27
> > **Thank you for the update**
> >
> > I thank the authors for making changes to the paper. It is much better now.
> >
> > The work overall is well done. But, it is disappointing that after such reward redistribution the optimal policy does not stay the same. This is the reason I am not increasing my score.

---

> > > ### Author Response · Authors · 2021-11-30
> > > **Thanks for your time and insightful comments that help us to improve our work.**
> > >
> > > Thanks for your time and insightful comments that help us to improve our work. Regarding the issue of $\pi^\star$-invariance, we would like to emphasize that this paper aims to develop a scalable and effective reward redistribution algorithm for practical purpose, which is demonstrated by our empirical results. It is a valuable future direction for us to consider how to maintain the $\pi^\star$-invariance property theoretically in RRD framework.

---

### Official Review · Reviewer_ySc8 · 2021-11-03

**Correctness:** 4
**Technical Novelty And Significance:** 3
**Empirical Novelty And Significance:** 3
**Recommendation:** 8
**Confidence:** 4

**Main Review:**

**Originality & significance**:
This paper addresses the important delayed reward problem in RL. The proposed RRD method is simple but somewhat novel and the interpretation of RRD as an interpolation between deterministic return decomposition and uniform credit assignment is inspiring. The empirical results show visible improvement of RRD over baseline methods and demonstrate the effectiveness of RRD.

**Quality**:
Overall the paper quality is good. Section 3 presents a nice analysis of RRD and connects it to existing methods. The empirical comparison to existing methods is adequate. The ablation study sheds light on understanding the effect of the number of samples per episode. I have two comments below on how to further enhance the empirical section.
1. It will be nice to gain some understanding of what the learned proxy reward function looks like. In many of these continuous control tasks, the main factor of the reward is the forward progress of the creature. It will be interesting to see if the learned proxy rewards are aligned with the forward progress.
2. Related to the previous point, the return in these continuous control tasks is mostly the sum of the forward progress at every step. Thus the return decomposition is fairly easy to learn. It will be interesting to see more empirical results in domains that are more challenging from a return decomposition perspective. For example, in video games (modified to have trajectory feedback), it is hard to credit every single move properly but certain critical events are clearly rewarding/punishing. The long episodes with sparse events in video games may also be a challenge for RRD as a small batch of randomly sampled state-action pairs may mostly contain uninformative trivial transitions but not many interesting events. It will be interesting to see if RRD can still outperform its deterministic counterpart in that case.

**Clarity**:
This paper is well written and easy to follow.

**Minor issues**:
In the first paragraph of Section 1, it says "Most standard RL frameworks... require the reward function to provide instant feedback for every step of environment transitions." I'm not sure if this statement is accurate. In theory, most RL algorithms do not make assumptions on the density of the rewards, though they may not perform well in practice due to sparse rewards. This sentence feels a bit overstated to me.

**Summary Of The Paper:**

This paper addresses the delayed reward problem in reinforcement learning (RL). The authors propose an algorithm, randomized return decomposition (RRD), that learns a proxy reward function to provide immediate reward feedback to the RL agent. Theoretical analysis shows that RRD is an interpolation between two existing methods, namely deterministic return decomposition and uniform credit assignment. RRD can be interpreted as a regularized version of the deterministic return decomposition and is a generalization of uniform credit assignment. Empirical results in the Mujoco continuous control benchmark show that RRD performs better than algorithms that are based on deterministic return decomposition and uniform credit assignment, and two other methods that learn auxiliary rewards for facilitating policy learning.

**Summary Of The Review:**

This paper addresses the important delayed reward problem in RL. The proposed method, RRD, is simple yet somewhat novel. The theoretical analysis builds connections to existing methods and the empirical study shows promises of RRD. Thus I think this paper is above the acceptance threshold.

---

> ### Author Response · Authors · 2021-11-23
> **Response to Reviewer ySc8**
>
> Thanks for the inspiring comments. We provide clarification to your questions and concerns as below. If our response does not fully address your concerns, please post additional questions and we will be happy to have further discussions.
>
> **Q1: It will be interesting to see if the learned proxy rewards are aligned with the forward progress.**
>
> In Appendix D, we visualize the correlation between the learned proxy rewards and the forward distance of the agent. We plot a scatter diagram, in which two axes correspond to the ground-truth agent motion and the assigned proxy reward, respectively. The plot shows that the learned proxy reward of RRD has a high correlation to the forward distance at the corresponding step.
>
> **Q2: It will be interesting to see more empirical results in domains that are more challenging from a return decomposition perspective, e.g., in video games.**
>
> In Appendix C, we add experiments on several Atari games. We modify the reward function to an episodic reward setting as we did in MuJoCo benchmark. These video games are more challenging than MuJoCo locomotion tasks, since (1) they have a longer task horizon and (2) the ground-truth per-step rewards are specified by complicated logic.
>
> We evaluate the performance of RRD built upon DQN and compare its performance against baseline algorithms. As shown in Figure 3, our method significantly outperforms IRCR and RUDDER. We will include more experiments on the Atari benchmark in the next revision.
>
> **Q3: In theory, most RL algorithms do not make assumptions on the density of the rewards, though they may not perform well in practice due to sparse rewards. This sentence feels a bit overstated to me.**
>
> We thank the reviewer for pointing out this writing flaw. The reviewer's understanding is correct. We would like to deliver a message that most algorithms prefer dense reward functions for better practical performance. In the rebuttal revision, we refine the discussion around this sentence to make the argument more rigorous.

---

### Official Review · Reviewer_qFZu · 2021-11-09

**Correctness:** 4
**Technical Novelty And Significance:** 3
**Empirical Novelty And Significance:** 3
**Recommendation:** 8
**Confidence:** 2

**Main Review:**

Strengths:
- Paper is very well written and flows well. The theory is well presented and discussed and is relatively easy to understand.
- The paper is well situated in the context of related work. Baselines are thoroughly discussed and extreme care is taken explain the differences between RRD and other alternatives.
- Experiments and ablations seem very thorough.

Weaknesses:
- The related work (section 5) seems relatively small. The paper would benefit from more text here.
- Experiments are constrained to Mujoco benchmark tasks. Would be interested in whether the authors have tried / intend to try the method on more such benchmarks.
- The graphs / figures might be more readable if some smoothing is introduced (and the smoothing factor is mentioned in the text). This is a matter of personal preference however so do not feel the need to do so.

**Summary Of The Paper:**

This paper targets how to decompose delayed reward signals obtained at the end of a trajectory to a more immediate form of reward feedback. The authors introduce an algorithm, "Reward Distribution vis Randomized Return Decomposition" that serves as a proxy reward function for episodic reinforcement learning. The authors showcase extensive experiments showcasing improved performance over relevant baselines on common Mujoco benchmarks. The main contribution is to convert long horizon delayed reward problems to more short length sequences that can be trained upon with mini-batch gradient descent.

**Summary Of The Review:**

The paper is well written and relatively easy to follow. It appears to introduce a novel algorithm for an important problem and should serve as an important alternative to training agents on long horizon delayed reward problems.

---

> ### Author Response · Authors · 2021-11-23
> **Response to Reviewer qFZu**
>
> Thanks for the inspiring comments. We provide clarification to your questions and concerns as below. If our response does not fully address your concerns, please post additional questions and we will be happy to have further discussions.
>
> **Q1: The related work (section 5) seems relatively small. The paper would benefit from more text here.**
>
> Due to the page limit, we summarize the most related papers in section 5 and include an enriched discussion in Appendix F. We make efforts to cover the related topics as much as we know. If we miss any relevant prior work, we would appreciate you can post further feedback and we will include them in the next revision.
>
> **Q2: Would be interested in whether the authors have tried / intend to try the method on more such benchmarks.**
>
> In Appendix C, we add experiments on several Atari games. We modify the reward function to an episodic reward setting as what we did in MuJoCo benchmark. These video games are more challenging than MuJoCo locomotion tasks, since (1) they have a longer task horizon and (2) the ground-truth per-step rewards are specified by complicated logic.
>
> We evaluate the performance of RRD built upon DQN and compare its performance against baseline algorithms. As shown in Figure 3, our method significantly outperforms IRCR and RUDDER. We will include more experiments on Atari benchmark in the next revision.
>
> **Q3: The graphs / figures might be more readable if some smoothing is introduced (and the smoothing factor is mentioned in the text).**
>
> We thank the reviewer for this suggestion. In this rebuttal revision, we perform two modifications on the plotted curves.
>
> - We smooth the curves by averaging over 10 most recent evaluation points. The evaluation points are set every $10^4$ timesteps in MuJoCo experiments. (In the initial submission, we smoothed the curves by averaging 5 evaluation points but forgot to mention it in the main text.)
> - We plotted the average performance instead of the median performance (suggested by reviewer 73tN).
>
> By the above modifications, the curves look nicer than those in the initial submission and the conclusions of experiments do not change.

---

### Decision · Program_Chairs · 2022-01-20

**Decision:**

Accept (Spotlight)

**Comment:**

Description of paper content:

The paper addresses the problem of credit assignment for delayed reward problems. Their method, Randomized Return Decomposition, learns a reward function that provides immediate reward. The algorithm works by randomly subsampling trajectories and predicting the empirical return by regression using a sum of rewards on the included states. The method is compared to a variety of existing methods on Mujoco problems in “episodic reward” settings, where the reward is zero except for the final step of the episode, where it is the sum of rewards from the original task. Theoretical argument suggests the method is an interpolation of return decomposition (regress based on all states, not a subsample) and uniform reward distribution (send episodic reward to all states equally). By regressing with a subset of states, the method reduces compute for longer problems and is suggested to be more scalable.

Summary of paper discussion:

The reviewers largely commended the simplicity of the method, the simplicity of the presentation, the novelty of the algorithm, and the quality of the empirical results. The negative reviewer maintained their initial review’s score on account of a bias introduced by the algorithm.